Accelerometer-measured physical activity is not associated with two-year weight change in African-origin adults from five diverse populations

Dugas Lara R. ldugas@luc.edu 1
Kliethermes Stephanie 2
Plange-Rhule Jacob 3
Tong Liping 1
Bovet Pascal 4 5
Forrester Terrence E. 6
Lambert Estelle V. 7
Schoeller Dale A. 8
Durazo-Arvizu Ramon A. 1
Shoham David A. 1
Cao Guichan 1
Brage Soren 9
Ekelund Ulf 9 10
Cooper Richard S. 1
Luke Amy 1
1 Public Health Sciences, Stritch School of Medicine, Loyola University Chicago , Maywood , IL , United States
2 Department of Orthopedics & Rehabilitation, University of Wisconsin, Madison , Madison , WI , United States
3 Department of Physiology, Kwame Nkrumah University of Science and Technology , Kumasi , Ghana
4 Institute of Social & Preventive Medicine, Lausanne University Hospital , Lausanne , VD , Switzerland
5 Ministry of Health , Victoria , Republic of Seychelles
6 Solutions for Developing Countries, University of West Indies, Mona , Kingston , Jamaica
7 Division of Exercise Science and Sports Medicine, Health Sciences, University of Cape Town , Cape Town , South Africa
8 Nutritional Sciences, University of Wisconsin, Madison , Madison , WI , United States
9 MRC Epidemiology Unit, University of Cambridge , Cambridge , United Kingdom
10 Department of Sport Medicine, Norwegion School of Sport Sciences , Oslo , Norway
Marusic Ana
Electronic publication date: 2017 Jan 19
Publication date: 2017
Volume: 5
Electronic Location ID: e2902
Received 2016 Sep 15; Accepted 2016 Dec 12
Copyright: ©2017 Dugas et al.
Copyright year: 2017
Copyright holder: Dugas et al.
License: This is an open access article distributed under the terms of the Creative Commons Attribution License, which permits unrestricted use, distribution, reproduction and adaptation in any medium and for any purpose provided that it is properly attributed. For attribution, the original author(s), title, publication source (PeerJ) and either DOI or URL of the article must be cited.
License URL: https://creativecommons.org/licenses/by/4.0/

Keywords: Obesity, Weight gain, Physical activity

Funding: National Institutes of Health 1R01DK80763 This study was funded by the National Institutes of Health (1R01DK80763). There was no additional external funding received for this study. The funders had no role in study design, data collection and analysis, decision to publish, or preparation of the manuscript.

==============================
Background

Increasing population-levels of physical activity (PA) is a controversial strategy for managing the obesity epidemic, given the conflicting evidence for weight loss from PA alone per se. We measured PA and weight change in a three-year prospective cohort study in young adults from five countries (Ghana, South Africa, Jamaica, Seychelles and USA).

Methods

A total of 1,944 men and women had baseline data, and at least 1 follow-up examination including measures of anthropometry (weight/BMI), and objective PA (accelerometer, 7-day) following the three-year study period. PA was explored as 1-minute bouts of moderate and vigorous PA (MVPA) as well as daily sedentary time.

Results

At baseline; Ghanaian and South African men had the lowest body weights (63.4 ± 9.5, 64.9 ± 11.8 kg, respectively) and men and women from the USA the highest (93.6 ± 25.9, 91.7 ± 23.4 kg, respectively). Prevalence of normal weight ranged from 85% in Ghanaian men to 29% in USA men and 52% in Ghanaian women to 15% in USA women. Over the two-year follow-up period, USA men and Jamaican women experienced the smallest yearly weight change rate (0.1 ± 3.3 kg/yr; −0.03 ± 3.0 kg/yr, respectively), compared to South African men and Ghanaian women greatest yearly change (0.6.0 ± 3.0 kg/yr; 1.22 ± 2.6 kg/yr, respectively). Mean yearly weight gain tended to be larger among normal weight participants at baseline than overweight/obese at baseline. Neither baseline MVPA nor sedentary time were associated with weight gain. Using multiple linear regression, only baseline weight, age and gender were significantly associated with weight gain.

Discussion

From our study it is not evident that higher volumes of PA alone are protective against future weight gain, and by deduction our data suggest that other environmental factors such as the food environment may have a more critical role.

Introduction

Increasing population-levels of physical activity (PA) is considered a strategy for managing the obesity epidemic but this is controversial, with both sides of the debate providing strong arguments (Hill, Peters & Blair, 2015; Luke & Cooper, 2013). Many experts argue that a decline in PA, particularly from occupation-related activities, is a key contributor to the current obesity epidemic (Archer et al., 2013; Blair, Archer & Hand, 2013; Hill & Peters, 2013; Popkin, 2001; Popkin, Adair & Ng, 2012; Popkin & Gordon-Larsen, 2004); however, prospective longitudinal studies employing objective measures of PA have not identified a meaningful relationship between weight gain and PA (Dugas et al., 2014; Ebersole et al., 2008; Luke et al., 2014; Swinburn, Sacks & Ravussin, 2009; Tataranni et al., 2003). The debate over the primary determinants of the worldwide obesity epidemic is more than simply academic; understanding obesity’s etiology is critical for informing policy and allocating scarce public health funding.

Most studies examining the relationship between longitudinal weight gain and PA have relied on self-report measures, typically questionnaires (Paul et al., 2014; Song et al., 2014). The nature of many common PA questionnaires lend themselves to significant over-reporting of daily PA levels (Atienza et al., 2011; Belcher et al., 2014; Schuna Jr et al., 2013; Tucker, Welk & Beyler, 2011; Tucker et al., 2015), as participants are asked to report domain-based PA levels (i.e., transport, occupation and leisure time), and inaccuracy can also arise because some questionnaires often request participants to only report PA when they engage in PA during periods of at least 10-mins. Due to the social desirability of portraying oneself as engaging in healthy behaviors, e.g., being physically active, survey participants often report higher than actual levels of healthful behaviors (Klesges et al., 2004; Phillips & Clancy, 1972), and this bias may differ according to sex, socio-economic status and other characteristics. This over-statement of daily PA was illustrated by Tucker et al. (Tucker, Welk & Beyler, 2011) using NHANES data, who found that 59.6% of USA adults reported that they met the USA Surgeon General guidelines (i.e., accumulated 150 min/week of moderate and vigorous activity); however, using objective activity monitoring data, only 8.2% overall of the same USA adults met the recommended guidelines. This also demonstrates the need to rely on objectively measured PA data to fully understand the role of PA in health.

The Modeling the Epidemiologic Transition Study (METS) is a prospective cohort study of weight change in 2,500 young adults aged 25–45 years of predominantly African descent from community-based samples in five countries at different levels of socio-economic development (Luke et al., 2012; Luke et al., 2014). The countries span the UN Human Development Index (HDI), thus representing a range of social and economic development, and include Ghana, South Africa, Jamaica, Seychelles and the USA. Participants were examined in local study clinics every year, which provide yearly measures of participant characteristics, health behavior, and objectively measured PA. The purpose of this study was to objectively assess whether PA measured at baseline is associated with subsequent two-year weight change in a cohort of diverse populations.

Materials and Methods

Sampling design and participant recruitment

Twenty-five hundred adults, ages 25–45, were enrolled in METS between January 2010 and December 2011. After the baseline exam, participants were asked to return to the clinic yearly for an additional two clinic exams: one-year follow-up and two-year follow-up. Participants were excluded if they self-reported that they were either HIV positive or pregnant at the time of recruitment. During the follow-up period, if a participant was pregnant or lactating, their weight data was not recorded during that year’s follow-up visit; only the health history was recorded. Participants were also asked about voluntary weight loss and other recreational drug use. One person was excluded from the USA cohort for intentionally losing 36 kg. For the current analysis, follow-up time varied between 1.32 years in Jamaican men to 1.98 years in Seychellois women (Table 1). Consequently, all analyses were adjusted for differing follow-up times and as described in the statistics section. A detailed description of the study protocol has been previously published (Luke et al., 2012). In brief, participants (approximately 50% of whom were female) were enrolled in each of five study sites: rural Ghana, urban South Africa, the Seychelles, urban Jamaica and metropolitan USA (Chicago). The participants were predominantly of African descent from each of the five countries. Study sites were selected to represent a broad range of social and economic development as defined by the UN HDI 2011: i.e., Ghana as a low-middle HDI country (HDI rank 135), South Africa as middle (123), Jamaica (80) and the Seychelles as high (52), and the US as a very high HDI country (4).

Table 1 Participants’ characteristics by site for men (N = 869) and women (N = 1, 027).

	Men (N= 888)	
	Ghana (N= 180)	RSA (N = 183)	Jamaica (N = 198)	Seychelles (N = 171)	USA (N = 156)	
Age (yr)	35.66 ± 6.43	33.79 ± 5.55	34.13 ± 5.98	36.62 ± 5.08	35.95 ± 6.40	
Baseline weight (kg)	63.35 ± 9.48	64.89 ± 11.79	73.71 ± 15.72	80.29 ± 16.21	93.16 ± 25.24	
Baseline normal weight (BMI < 25 kg/m2)	82.69%	79.78%	67.22%	41.92%	28.65%	
Weight change (kg)	0.69 ± 3.15	0.76 ± 4.67	0.20 ± 3.90	0.81 ± 5.22	0.11 ± 5.99	
Time change (yr)	1.70 ± 0.48	1.61 ± 0.62	1.32 ± 0.46	1.98 ± 0.27	1.54 ± 0.54	
Weight change rate (kg/yr)	0.48 ± 2.16	0.50 ± 3.44	0.21 ± 3.31	0.42 ± 2.61	−0.08 ± 4.47	
MVPA (min/day 10-min bout)	24.68 ± 18.74	31.66 ± 27.79	12.95 ± 16.62	16.67 ± 17.11	18.88 ± 29.45	
MVPA (min/day 1-min bout)	51.54 ± 26.25	56.26 ± 34.46	31.37 ± 24.65	36.34 ± 24.43	33.45 ± 34.29	
Light (min/day 10-min bout)	144.20 ± 81.07	76.88 ± 55.40	106.00 ± 65.00	129.84 ± 87.12	117.67 ± 80.00	
Light (min/day 1-min bout)	253.65 ± 77.51	189.11 ± 69.33	219.30 ± 70.76	232.11 ± 77.97	235.69 ± 82.32	
Sedentary (min/day 10-min bout)	54.08 ± 40.15	52.00 ± 32.99	79.92 ± 53.05	68.82 ± 51.13	56.66 ± 36.45	
Sedentary (min/day 1-min bout)	228.97 ± 47.56	241.37 ± 45.16	267.01 ± 59.89	212.67 ± 57.92	249.71 ± 56.15	
Meets US PA guidelines (Yes)	74.36%	76.50%	42.78%	55.56%	38.01%	
	Women (N = 1,055)	
	Ghana (N = 186)	SA (N = 234)	Jamaica (N = 215)	Seychelles (N = 197)	USA (N = 223)	
Age (yr)	34.87 ± 6.64	33.05 ± 5.98	34.92 ± 6.24	35.78 ± 6.02	35.77 ± 6.01	
Baseline weight (kg)	63.28 ± 12.67	82.78 ± 21.87	80.04 ± 19.16	70.65 ± 16.11	91.94 ± 23.35	
Baseline normal weight (BMI < 25 kg/m2)	51.57%	19.66%	26.88%	40.00%	13.20%	
Weight change (kg)	1.94 ± 4.10	2.15 ± 6.82	−0.11 ± 3.72	2.33 ± 4.27	0.08 ± 6.60	
Time change (yr)	1.72 ± 0.49	1.96 ± 0.39	1.48 ± 0.51	1.99 ± 0.24	1.64 ± 0.53	
Weight change rate (kg/yr)	1.17 ± 2.69	0.98 ± 3.40	−0.03 ± 2.79	1.15 ± 2.17	0.002 ± 4.27	
MVPA (min/day 10-min bout)	11.79 ± 12.70	10.17 ± 10.14	8.76 ± 10.07	11.00 ± 10.33	6.56 ± 13.50	
MVPA (min/day 1-min bout)	28.32 ± 18.93	22.31 ± 15.41	21.37 ± 16.78	24.44 ± 14.62	15.77 ± 19.31	
Light (min/day 10-min bout)	165.07 ± 72.60	93.24 ± 51.33	112.39 ± 60.48	126.82 ± 78.61	108.35 ± 75.74	
Light (min/day 1-min bout)	267.39 ± 69.15	200.04 ± 57.36	221.45 ± 63.61	229.54 ± 73.61	223.29 ± 77.01	
Sedentary (min/day 10-min bout)	46.83 ± 27.62	61.02 ± 37.70	57.43 ± 34.46	52.56 ± 33.95	52.36 ± 35.66	
Sedentary (min/day 1-min bout)	222.69 ± 41.70	246.48 ± 44.49	243.93 ± 45.42	198.74 ± 50.10	240.49 ± 57.39	
Meets USA PA guidelines (Yes)	38.12%	24.79%	28.49%	29.30%	14.72%	

METS was approved by the Institutional Review Board (IRB) of Loyola University Chicago, IL, USA; the Committee on Human Research Publication and Ethics of Kwame Nkrumah University of Science and Technology, Kumasi, Ghana; the Health Sciences Faculty Research Ethics Committee of the University of Cape Town, South Africa; the Board for Ethics and Clinical Research of the University of Lausanne, Switzerland; the Research Ethics Committee of the Ministry of Health in Seychelles; the Ethics Committee of the University of the West Indies, Kingston, Jamaica; and the Health Sciences IRB of the University of Wisconsin, Madison, WI, USA. Written informed consent was obtained from all participants.

Measurements

Yearly measurements were made at outpatient clinics located in the respective communities. Weight and height measurements were made on all participants while wearing light clothing and no shoes and using the same model of equipment, which was calibrated using the same protocol at each locale. Weight was measured to the nearest 0.1 kg using the same standard balance at all five sites and during the entire study period (Seca 770; Seca, Hamburg, Germany). Height was measured to the nearest 0.1 cm using a stadiometer (e.g., Invicta Stadiometer; Invicta, London, UK). Waist circumference was measured to the nearest 0.1 cm at the umbilicus and hip at the point of maximum extension of the buttocks. Body mass index (BMI) was calculated as kg/m2.

Body composition was estimated by bioelectrical impedance analysis (BIA) with the use of single-frequency (50 kHz) impedance analyzer (model BIA 101Q; RJL Systems, Clinton Township, MI). Fat-free mass (FFM) and fat mass (FM) were estimated from measured resistance by using an equation validated in the METS cohorts (Luke et al., 1997).

PA was assessed using Actical accelerometers (Phillips Respironics, Bend, OR, USA) as previously described (Dugas et al., 2014; Luke et al., 2014). Briefly, the monitor was worn at the waist, over an 8-day period encompassing the partial first and last days, including during sleep. We assessed activity conducted between the hours of 7 am–11 pm daily. Raw data downloaded from the accelerometers were first passed through a National Cancer Institute (2014) designed to infer non-wear time from 90 or more minutes of continuous zero activity counts. A valid day of PA monitoring was defined as one having 10 or more hours of wear time and participant files were included if they contained ≥4 or more valid days. Using the same protocol employed by the National Center for Health Statistics for the analysis of accelerometry data in the continuous National Health and Nutrition Examination Survey (Troiano et al., 2008), minutes defined as comprising sedentary, moderate, vigorous or moderate plus vigorous activity are presented as the total time in minutes (min) per day accumulated in either 1- or 10-min intervals and this was averaged for the number of days with valid accelerometry data.

Statistical analysis

Data management was centralized at Loyola University Chicago. All data forms and questionnaires were scanned at each study site and, along with electronic Actical data files, were sent via secure FTP (Bitvise Tunnelier, Version 4.40, http://www.bitvise.com/tunnelier) to the data manager at the coordinating center.

Descriptive statistics were used to summarize the characteristics of participants in each of the five study sites, including descriptive characteristics of the physical activity variables. Because the yearly follow up periods were not identical in the different sites or within sites, all analyses on weight change were adjusted for the different follow-up periods. Univariate analyses were conducted to determine partial correlation coefficients between parameters of PA and two-year weight change, after adjustment for age and follow up duration, and accounting for multiple observations for each individual (Hill, Peters & Blair, 2015). Partial correlation coefficients between weight change and PA are also presented separately among persons who gained weight and among persons who lost weight during follow up. For the purpose of our analysis, yearly weight change (kg/yr) was defined as the difference between the first and last recorded weight measure over the follow-up period, while weight loss and weight gain were defined simply by whether the final weight measurement either exceeded or was lower than the baseline starting weight (Table 2).

Table 2 Yearly weight change (kg/yr) for BMI status and meeting PA guidelines at baseline (MVPA ≥ 30 min/day) for men and women, by site (means ±  standard deviation).

		Ghana	RSA	Jamaica	Seychelles	USA	
Men (N = 888)	Normal weight (BMI < 25 kg/m2)	N = 129	N = 146	N = 121	N = 83	N = 49	
0.58 ± 2.09	0.64 ± 3.12	0.14 ± 2.26	0.56 ± 1.88	0.23 ± 2.22	
Overweight (BMI ≥ 25 ≤ 30 kg/m2)	N = 24	N = 30	N = 41	N = 73	N = 53	
0.28 ± 2.12	−0.20 ± 4.68	0.44 ± 3.67	0.66 ± 2.73	−0.27 ± 2.94	
Obese (BMI ≥ 25 kg/m2)	N = 3	N = 7	N = 18	N = 42	N = 69	
−2.06 ± 4.67	0.62 ± 3.95	0.21 ± 6.99	−0.28 ± 3.44	−0.16 ± 6.31	
Women (N = 1, 055)	Normal weight (BMI < 25 kg/m2)	N = 115	N = 46	N = 50	N = 86	N = 26	
1.37 ± 2.53	1.00 ± 3.30	0.23 ± 1.95	1.29 ± 1.88	1.15 ± 3.14	
Overweight (BMI ≥ 25 ≤ 30 kg/m)2	N = 75	N = 57	N = 41	N = 68	N = 41	
1.08 ± 3.02	1.35 ± 3.41	0.08 ± 3.49	1.30 ± 2.06	0.77 ± 4.08	
Obese (BMI ≥ 25 kg/m2)	N = 33	N = 131	N = 95	N = 61	N = 130	
0.72 ± 2.45	0.81 ± 3.44	−0.21 ± 2.84	0.78 ± 2.61	−0.47 ± 4.47	
Men (N = 888)	Meets USA PA guidelines	N = 116	N = 140	N = 77	N = 110	N = 65	
0.54 ± 2.21	0.57 ± 3.57	0.59 ± 4.35	0.55 ± 2.18	0.25 ± 5.05	
Does not meet PA guidelines	N = 40	N = 43	N = 103	N = 88	N = 106	
0.30 ± 2.02	0.28 ± 3.02	−0.07 ± 2.77	0.25 ± 3.06	−0.29 ± 4.09	
Women (N = 1, 056)	Meets USA PA guidelines	N = 85	N = 58	N = 53	N = 63	N = 29	
1.26 ± 2.50	1.75 ± 3.09	−0.03 ± 3.13	1.44 ± 2.00	−0.35 ± 5.05	
Does not meet PA guidelines	N = 138	N = 176	N = 133	N = 152	N = 169	
1.12 ± 2.81	0.73 ± 3.47	−0.02 ± 2.65	1.03 ± 2.23	0.06 ± 4.14	
Notes.

Adjusted for age and time.

Multilevel models

Because the follow-up time was different between participants and sites, and some participants only had one follow-up measurement, we used the multi-level random slope and intercept model, which allowed us to use all available data points, to examine the associations between weight PA variables, including sedentary time. Briefly, for Level 1; an individual’s weight (kg) from each examination was regressed on follow-up time. Secondly, level 2 models the potential dependence of the individual baseline weight (random intercept from level 1) and rate of weight change (kg/yr, random slope of level 1) as a function of the other covariates. The multi-level model is as follows: (1) Level1:Weightij=β0i+β1iTij+εij

(2) Level2:β0i=α0+α1BaseWeighti+α2Agei+α3Genderi+α4Sitei+α5PhysicalActivityi+ξ0i

(3) β1i=γ0+γ1ghanai+γ2rsai+γ3jamaicai+γ4seychellesi+γ5PhysicalActivityi+ξ1i,

where i = 1, 2…N and j = 1, 2. The effect of the PA variables on weight change is measured by the interaction coefficient γ5, and the average effect of the PA variables on weight change estimate by α5. We have previously used this multi-level approach (Luke et al., 2009), which utilizes all available data, including incomplete data, and the within- and between-person variability, while adjusting for potential confounders such as age and baseline weight. All analyses were conducted using SAS (SAS Institute, Cary, NC USA, version 9.4) and a conventional two-sided 5% alpha error was used to denote statistical significance.

Figure 1 The correlation coefficient (adjusted for age and follow up time) between weight change and MVPA (1-min bouts) for men is 0.03459 (n = 869, p = 0.3091).

Results

Subject characteristics

Baseline measures where completed in 2010 and 2011 and follow-up yearly exams during 2011/2012 (follow-up 1) and 2012/2013 (follow-up 2). At baseline, a total of 2,506 participants were enrolled from the 5 sites over the enrollment period. Two thousand and sixty eight participants returned for a 1 year follow-up, Ghana (n = 417), South Africa (n = 414), Jamaica (n = 399), Seychelles (n = 481) and the USA (n = 357). While a total of 1,408 participants returned for a second follow-up visit, Ghana (n = 289), South Africa (n = 322), Jamaica (n = 136), Seychelles (n = 456) and finally, the USA (n = 205). The final sample therefore included a total of 888 men and 1,056 women with at least complete baseline measurements and 1 follow-up clinic visit (Table 1). We did exclude one participant from the USA who intentionally lost 36 kg after the baseline visit.

The mean age among the men and women were 34.9 ± 6.16 yr and ranged from 33.7 ± 5.6 yr in the South African men to 36.6 ± 5.1 yr in the Seychellois men. Among men and women, mean time between the baseline visit and 2year follow-up visit ranged from 1.32 ± 0.46 years among the Jamaican men and 1.98 ± 0.27 years among the Seychellois women. As a result, all of our statistical modeling was adjusted for both age and follow-up time differences.

The mean baseline weights were different among the 5 sites, with the lowest weights being captured among men from Ghana and South Africa (63.4 ± 9.5 and 64.9 ± 11.8 kg, respectively) and highest among the USA men (93.6 ± 25.9 kg). Among the women, Ghanaians weighed less than USA women (63.3 ± 12.7 vs. 91.7 ± 23.4 kg). Figures 1A and 1B present the weight change and PA data by site. After the two-year follow-up period, surprisingly USA men experienced the smallest yearly weight change 0.05 ± 3.27 kg/yr, while among the women, the Jamaicans actually lost on average −003 ± 2.79 kg/yr. In both men and women, yearly weight change was always greater among participants categorized as normal weight (BMI < 25 kg/m2) at baseline. For example, normal weight men from Seychellois and Ghana experienced a yearly weight gain between 0.56–0.58 kg/yr, respectively, compared to their obese counterparts who actually experienced a yearly weight loss between −0.28–2.06 kg/yr, respectively (Table 2). Similarly, among the women, yearly weight gain was greatest among the normal and overweight participants, than among the obese participants, ranging from 0.23 ± 1.95 kg/yr in the Jamaicans up to 1.37 ± 2.53 kg/yr among the Ghanaian women.

Table 3 presents the absolute weight change data across the entire follow-up period, by site and sex, stratifying for either overall weight gain or weight loss. The number of men gaining weight (end weight > start weight) ranged from 40% among the Jamaicans to 54% among the Seychellois. The absolute weight gain varied from 3.2 ± 2.4 kg to 4.5 ± 3.5 kg. Among the women, the number of participants gaining weight varied widely from 44% of the Jamaicans, who gained on average 3.1 ± 1.9 kg up to 77% of the South African women, who gained 6.4 ± 5.3 kg.

Table 3 Overall weight loss/gain by site for men and women (means ±  standard deviation).

		Ghana	RSA	Jamaica	Seychelles	USA	
Men (N = 874)	Weight loss	N = 51	N = 75	N = 74	N = 68	N = 75	
−2.42 ± 1.87	−3.21 ± 2.74	−2.90 ± 3.03	−3.64 ± 5.17	−4.44 ± 4.56	
Weight gain	N = 72	N = 85	N = 71	N = 105	N = 79	
3.21 ± 2.36	4.49 ± 3.50	3.55 ± 2.78	3.88 ± 3.43	4.47 ± 4.39	
Women (N = 1, 048)	Weight loss	N = 57	N = 81	N = 78	N = 44	N = 77	
−2.86 ± 2.07	−4.57 ± 3.33	−3.43 ± 2.79	−2.96 ± 2.31	−5.87 ± 5.58	
Weight gain	N = 146	N = 137	N = 81	N = 148	N = 101	
4.04 ± 3.17	6.37 ± 5.28	3.06 ± 1.85	4.27 ± 3.46	4.65 ± 3.66	

Accelerometer PA at baseline, measured as either 10-min or 1-min bouts (Table 1), was greatest among the South African men, who accumulated almost 1 hr/day of MPVA in 1-min bouts and among the women, greatest in the Ghanaian women, who accumulated almost 30 min/day in 1-min MVPA bouts. Sedentary activity, also measured as either 10-min or 1-min bouts, was greatest among the Jamaican men (266.9 ± 60.2 min/day in 1-min bouts) and highest among the South African women (247.2 ± 44.7 min/day in 1-min bouts).

We categorized participants at each site by whether they met the USA Surgeon General guidelines for PA, i.e., 30 min/day on most days of the week, using the 1-min MVPA data. Among the men, over 76% of Ghanaians (N = 158) and South Africans (N = 183) met the guidelines, compared to only 44% of USA men (N = 109). Far fewer women met this guideline, with only 44% of Ghanaian women (N = 130) compared to about 20% of USA women (N = 50).

Surprisingly, the total weight gain at every site, were actually greater among participants who met the PA guidelines (Table 2) than among their site counterparts who did not meet the PA guidelines. For example among USA men who met the PA guidelines, yearly weight change was 0.25 ± 5.05 kg/y compared to a loss of 0.29 ± 4.09 kg/yr among those not meeting the guideline. Similarly, among the women, at each site, weight gain was less among the participants not meeting the PA guidelines, for example among South Africa women meeting the PA guidelines the yearly weight gain was 1.8 ± 3.1 kg/yr vs. 0.7 ± 3.5 kg/yr. USA women were the only exception, where women meeting the guideline lost 0.4 ± 5.1 kg/yr, compared to an average weight increase of 0.1 ± 4.1 kg/yr in those failing to meet the guideline.

At baseline, overweight Ghanaians and South Africans were 0.49 (95%CI [0.32–0.75], p = 0.001) and 0.48 (95% CI [0.29–0.77], p = 0.003) times as likely to meet the PA guidelines at baseline compared to normal weight participants. Among the obese, Ghanaians, South Africans and USA participants were 71%, 78% and 62% less likely to meet the PA guidelines (p < 0.001 for all) at baseline.

Relationship between weight change, and baseline physical activity and sedentary behavior

Tables 4A–4C and Tables 5A–5C present the partial correlation coefficients between weight change and MVPA (Tables 4A–4C), and sedentary behavior (Tables 5A–5C). Because overall 10-min bout MVPA was so low, we chose to model our exploratory statistics using 1-min bout MVPA only. We adjusted all analyses for age and length of follow-up and restricted the analyses to each site to account for differences in site weight change.

Table 4 Correlations between weight change, weight gain and weight loss and baseline MVPA (1-min bouts) by site and gender.

		Ghana	RSA	Jamaica	Seychelles	USA	
(A) Weight change velocity	
Men	Correlation	−0.044	0.039	0.056	−0.017	0.046	
p-value	0.58	0.60	0.46	0.81	0.55	
N	156	183	180	198	171	
Women	Correlation	−0.019	0.048	0.031	0.153	−0.084	
p-value	0.78	0.47	0.67	0.03	0.24	
N	223	234	186	215	197	
(B) Weight gain (among participants who gained weight)	
Men	Correlation	−0.022	−0.039	0.012	−0.032	0.087	
p-value	0.84	0.70	0.92	0.71	0.41	
N	90	99	77	137	90	
Women	Correlation	−0.080	−0.021	0.004	0.091	0.064	
p-value	0.32	0.79	0.97	0.24	0.50	
N	158	157	93	167	115	
(C) Weight loss (among participants who lost weight)	
Men	Correlation	−0.015	0.092	0.055	−0.021	0.264	
p-value	0.91	0.39	0.63	0.85	0.02	
N	59	90	79	89	82	
Women	Correlation	0.023	0.070	−0.087	0.177	−0.013	
p-value	0.84	0.48	0.42	0.15	0.91	
N	78	105	88	68	86	
Notes.

Adjusted for age and time.

Table 5 Correlations between weight change, weight gain and weight loss and baseline sedentary time (1-min bouts) by site and gender.

		Ghana	RSA	Jamaica	Seychelles	USA	
(A) Weight change	
Men	Correlation	0.188	0.054	−0.059	−0.034	−0.003	
p-value	0.02	0.47	0.43	0.97	0.97	
N	156	183	180	198	171	
Women	Correlation	−0.140	0.025	0.008	0.014	0.044	
p-value	0.04	0.70	0.91	0.84	0.54	
N	223	234	186	215	197	
(B) Weight gain (among participants who gained weight)	
Men	Correlation	0.011	−0.001	0.001	−0.014	−0.043	
p-value	0.92	0.99	0.188	0.87	0.69	
N	90	99	99	137	90	
Women	Correlation	−0.126	0.000	−0.021	0.167	0.003	
p-value	0.11	0.99	0.84	0.03	0.97	
N	158	157	93	167	115	
(C) Weight loss (among participants who lost weight)	
Men	Correlation	0.021	0.042	−0.047	−0.001	0.001	
p-value	0.87	0.69	0.68	0.99	0.99	
N	59	90	79	89	82	
Women	Correlation	−0.005	0.214	0.009	−0.031	0.339	
p-value	0.9	0.03	0.93	0.80	0.001	
N	78	105	88	68	86	
Notes.

Adjusted for age and time.

Among the men, overall weight change did not correlate with 1-min bout MVPA at any of the sites. Because this was not the expected relationship, we also performed this analysis on only the participants who either gained or lost weight, but this also did not produce any statistically significant associations. For example, among the participants who lost weight, only women from the Seychelles had a significant relationship for 1-min bout MVPA and weight loss (r = 0.326, p = 0.015), while among participants who gained weight, the only significant relationship between PA and weight gain was among the USA women who actually had a positive result (r = 0.218, p = 0.026), indicating higher PA levels resulted in greater weight gain.

Sedentary behavior, in recent years, has been thought to be a key component contributing to an increased risk for overall morbidity and mortality, independent of daily PA. Therefore, we also examined the relationship between sedentary time at baseline and prospective weight change during the two-year follow up period (Tables 5A–5C). Weight change was negatively associated with sedentary time at baseline, only among the Ghanaian women (r =  − 0.147, p = 0.029), i.e., weight gain was less among those with greater sedentary time at baseline. We did not find any significant relationships between either weight gain or weight loss and baseline sedentary time among any of the sites (Figs. 2A and 2B). Furthermore, we found that the nature of the relationship between the two variables was inconsistent, with some relationships being positive, while others negative.

Figure 2 The correlation coefficient (adjusted for age and follow up time) between weight change and MVPA (1-min bouts) for women is 0.05997 (n = 1, 027, p = 0.0549).

Multi-level models

Table 6 presents the parameter estimates for weight change using baseline MVPA (model 1), baseline light activity (model 2) and baseline sedentary activity (model 3), all based on 1-min bouts, and with analyses adjusted for follow up time and age. In model 1, it can be seen that baseline MVPA is not significantly associated with prospective weight, only the interaction term between follow-up time and the South African site, where each year of follow-up time is associated with an increase in weight that is on average 0.96 kg greater than the USA (p = 0.02). This outcome was similarly found for model 2 (light activity and model 3, sedentary activity. In addition, among the 3 models; baseline weight, age and gender were all significantly associated with weight.

Table 6 Parameter estimates from the mixed-effects models for weight, using baseline MVPA (model 1), light (model 2) and sedentary activity (model 3).

Variable	Model 1: MVPA (1-min bouts)	Model 2: Light activity (1-min bouts)	Model 3: Sedentary activity (1-min bouts)	
	Estimate	95% CI	p-value	Estimate	95% CI	p-value	Estimate	95% CI	p-value	
Intercept	4.288	(2.688–5.887)	<0.001	4.448	(2.56, 6.33)	<0.001	3.995	(1.766, 6.224)	<0.001	
Baseline weight (kg)	0.986	(0.976–0.995)	<0.001	0.98	(0.975, 0.993)	<0.001	0.985	(0.976, 0.994)	<0.001	
Age (y)	−0.059	(−0.088, −0.0316)	<0.001	−0.062	(−0.090, −0.034)	<0.001	−0.06	(−0.089, −0.032)	<0.001	
Men	−0.45	(−0.821, −0.078)	0.02	−0.653	(−0.996, −0.301)	<0.001	−0.665	(−1.009, −0.320)	<0.001	
Site			0.29			0.34			0.28	
Ghana	−0.569	(−1.75, 0.611)	0.34	−0.528	(−1.705, 0.649)	0.38	−0.447	(−1.621, 0.728)	0.46	
RSA	−0.924	(−2.02, 0.168)	0.1	−0.874	(−1.959, 0.211)	0.11	−0.877	(−1.950, 0.196)	0.11	
Jamaica	−0.184	(−1.546, 1.179)	0.79	−0.173	(−1.531, 1.184)	0.8	−0.199	(−1.557, 1.160)	0.77	
Seychelles	−0.008	(−1.100, 1.085)	0.99	−0.019	(−1.104, 1.067)	0.97	0.087	(−1.032, 1.207)	0.88	
Activity estimate (1-min bout)	0.0004	(−0.013, 0.013)	0.95	0.0003	(−0.004, 0.005)	0.88	0.002	(−0.005, 0.008)	0.61	
Follow-up time (mo)	−0.201	(−0.905, 0.502)	0.57	0.176	(−0.855, 1.207)	0.74	0.036	(−1.376, 1.449)	0.96	
Follow-up time*site			0.06			0.052			0.05	
Ghana	0.644	(−0.247, 1.535)	0.16	0.632	(−0.253, 1.517)	0.16	0.586	(−0.296, 1.469)	0.19	
RSA	0.961	(0.124, 1.798)	0.02	0.94	(0.112, 1.768)	0.03	0.966	(0.145, 1.787)	0.02	
Jamaica	−0.329	(−1.42, 0.760)	0.55	−0.35	(−1.435, 0.736)	0.53	−0.337	(−1.424, 0.749)	0.54	
Seychelles	0.54	(−0.281, 1.361)	0.2	0.576	(−0.238, 1.390)	0.17	0.556	(−0.286, 1.398)	0.20	
MVPA*follow-up time	0.001	(−0.009, 0.011)	0.83	−0.001	(−0.005, 0.002)	0.46	−0.0009	(−0.006, 0.004)	0.77	

Discussion

The first finding is that body weight increased during the two year follow up period in both men and women, except among women in Jamaica. Second, there were significant differences in the two-year weight change according to sex and country, with a trend towards larger weight gain in countries with lower HDI. Third, weight gain tended to be lower among the obese participants than the normal weight participants, in all sites and sexes, and fourth, PA at baseline was not associated with two year follow up weight change.

Previously we reported weight change data from a similar international study comparing adults residing in either the USA, Jamaica and Nigeria over a four-year period between 1995–1999 (Durazo-Arvizu et al., 2008). Time-adjusted weight gain was highest among the Jamaicans (1.37 kg/yr), compared to USA (0.52 kg/yr) and Nigerians (0.31 kg/yr). Fifteen years ago Jamaica represented a country under-going rapid transition as a result of accelerating cultural and behavioral shifts. In our current data set, South Africa and Ghana represent countries under-going more rapid transition, and again we find similar results, whereby countries undergoing more rapid transition, as reflected by their changing HDI, experience greater time-adjusted yearly weight gain rates. The yearly weight gain rates for Ghana and South Africa were: 1.41 and 1.54 kg/yr respectively, compared to Jamaica, and the USA 0.05, and 0.09 kg/yr, respectively. The exception were the Seychelles, where the rate of weight change is 1.6 kg/yr.

Secondly, we found an overall lack of association between the PA attributes (i.e., MPVA and sedentary time) at baseline and prospective weight gain. Likewise, within each site, we surprisingly found that participants meeting PA guidelines at baseline, i.e., accumulating ≥30 min/day, experienced higher rates of yearly weight gain, compared to those not meeting PA guidelines. This is comparable to other large prospective studies such as the European Prospective Investigation into Cancer and Nutrition (EPIC, (Ekelund et al., 2011)) and the Women’s Health Study (Lee et al., 2010) and has several public health policy implications for the obesity epidemic. Indeed, the achievement of higher volumes of daily PA continues to be presented as a possible global solution and management tool for the epidemic (WHO, 2010; U.S. Department of Health & Human Services, 2010; Office of Disease Prevention and Health Promotion, 2012a; Office of Disease Prevention and Health Promotion, 2012b; National Physical Activity Plan, http://www.physicalactivityplan.org/index.html; WHO, 2013; Office of Disease Prevention and Health Promotion, 2013). From our study with 1,944 young adults, higher volumes of PA at baseline were not protective against future weight gain, suggesting that other environmental factors, such as the food environment, may have a more pivotal role in weight gain. Recent evidence from the United States-based Coronary Artery Risk Development in Young Adults (CARDIA) cohort (Richardson et al., 2015) using 20 years of BMI follow-up data, found significant effects for neighborhood fast food restaurants, such that BMI increases were associated with a higher consumption of an obesogenic fast food-type diet. This was after controlling for effects of SES and PA (Richardson et al., 2015). Interestingly, PA among the young adults decreased from baseline to the first follow-up (seven years), but then remained stable during the subsequent three follow-up measurements (13 years), while BMI continued to increase.

Importantly, this is not to say, that PA per se is not important for overall achievement of health such as the prevention or delay of diabetes and cardiovascular disease, which is undisputed (Glenn et al., 2015; Lin et al., 2015; Long et al., 2015), but that its role in the prevention of population level weight gain may be overstated.

Notably, yearly weight change rate, within each site, appeared to be smallest among the participants characterized as either overweight or obese at baseline, and in many instances overweight or obese participants actually experienced a negative yearly weight change rate. Overweight and obese men had a negative yearly weight change rate (−0.27 kg/yr and −0.16 kg/yr, respectively). A similar trend were captured among the women, where normal weight participants experienced greater yearly weight change rates, compared to their overweight and obese counterparts (Table 3).

Our data add to a growing body of literature reporting similar findings (Ekelund et al., 2011; Lee et al., 2010; Luke et al., 2014; Luke et al., 2009; Luke et al., 2006; Moholdt et al., 2014; Pontzer et al., 2012). Among participants in the EPIC study, baseline PA was not associated with prospective weight gain (Ekelund et al., 2011). Further, there were differences between younger (<50 yr) and older normal weight adults (≥50 yr) in the relationship between baseline PA and prospective weight gain (Ekelund et al., 2011). Lee et al. (2010) similarly found in 34,079 women (54.2 yr) no difference in weight gain (2.6 kg) comparing participants either meeting or not meeting the USA PA guidelines (≥150 min/week moderate PA), only in those meeting higher volume IOM PA guidelines (≥420 min/week moderate PA) over a period of 13 years. The measurement periods were separated into four measurement periods and examining changes in PA during the measurement period did also not result in differences in weight gain between the groups meeting or not meeting USA PA guidelines. Reported weight gain differences were only found among normal weight women (BMI<25 kg/m2) at baseline, who met the higher volume IOM PA guidelines, gaining less than 2.3 kg over the 13 year measurement period.

An important consideration is the direction of the PA-weight change relationship, i.e., does low PA contribute to greater future weight gain or does weight gain lead to future low PA levels? Golubic et al. (2013) found the second option to be true, namely weight gain is causal for low PA. Compared to participants who remained weight stable after 10 years of follow-up, those participants gaining between 0.5–2 kg were 1.26 times as likely to be inactive and almost twice as likely to be inactive if they gained more than 2 kg. While we do not present the longitudinal PA outcomes in this paper, we did explore the cross-sectional associations between meeting USA PA guidelines and BMI at baseline. Ghanaians, South Africans and Americans were significantly less likely to meet the PA guidelines if they were either overweight or obese. For example, obese Americans were 62% as likely to meet PA guidelines compared to normal weight Americans. While we cannot show any causality using a cross-sectional analysis, future investigations exploring the direction of the PA-weight gain relationship is a critical piece of this international debate, with significant public health implications. If weight gain or being overweight/obese precedes low PA levels, the public health message should reflect this.

We cannot ignore the wide ranging methodologies employed in studies exploring the relationship between PA and weight gain, or vice versa. Methodological and/or instrumentation differences are important factors in trying to compare the large body of evidence examining the relationship between weight gain and PA levels. Whereas data from studies such as ours employing objectively measured PA tend to show little or no correlation between PA and weight change, studies using self-report tend to show an association between PA and weight gain (Paul et al., 2014; Song et al., 2014). Self-report data may in fact express social desirability and be biased toward healthy behaviors, which may ultimately be associated with smaller weight gain. Some studies investigating the association between PA and eating behavior find that self-report PA is associated with increased eating self-regulation (Andrade et al., 2010; Martins, Morgan & Truby, 2008). Assessment of PA based on self-report is highly variable in different studies reflecting a number of problems with measurement of self-reported PA. A recent study comparing self-report data in USA adults participating in three surveillance systems (NHIS, NHANES and BRFSS) found considerable variation among the self-report PA variables, ranging from population PA levels of 30% (NHIS) up to 48% in BRFSS (Carlson et al., 2009). Tucker, Welk & Beyler (2011), found that USA adults may over report their PA by as much as 50%, with >60% of USA adults report that they meet the USA surgeon general guidelines (≥150 min/week of MVPA); however, using objective activity monitoring data, only 8% actually meet the same PA guidelines.

Our study has some limitations. While this population-based cohort is representative of the local communities, the current participants may not be entirely representative of the countries in which they reside and as such caution should be taken when interpreting the results across the human development spectrum. Secondly, as a result of the ongoing nature of this study, measurements were completed at different times of the calendar year/seasons at the five different sites. However, in order to reduce measurement errors, we provided the same brand/model of calibrated equipment items to all the research sites, and ensured their proper working order at all times. We also feel confident in the robustness of our data as a result of exploring potential seasonal/country differences and have found none to date (Sani et al., 2015).

Conclusion

In conclusion, in this prospective cohort study in young adults from five countries, representing different stages of epidemiologic transition, baseline PA was not associated with future weight gain after a two-year follow up. Further, among the different countries, we found that weight change patterns were different and could not simply be accounted for by level of development using a UN indicator. By deduction; our data suggest that other environmental factors that influence food consumption may be a more fertile field for public research and health intervention.

Supplemental Information

Data S1 Raw data

Click here for additional data file.

The authors would like to acknowledge the site-specific clinic staff members as well as the 2,500 participants.

Additional Information and Declarations

Competing Interests

Author Contributions

Human Ethics

Data Availability

Richard S. Cooper is an Academic Editor for PeerJ.

Lara R. Dugas conceived and designed the experiments, performed the experiments, analyzed the data, wrote the paper, prepared figures and/or tables, reviewed drafts of the paper.

Stephanie Kliethermes and Liping Tong analyzed the data, wrote the paper, prepared figures and/or tables, reviewed drafts of the paper.

Jacob Plange-Rhule, Pascal Bovet, Terrence E. Forrester, Estelle V. Lambert and Amy Luke conceived and designed the experiments, performed the experiments, contributed reagents/materials/analysis tools, wrote the paper, reviewed drafts of the paper.

Dale A. Schoeller contributed reagents/materials/analysis tools, wrote the paper, reviewed drafts of the paper.

Ramon A. Durazo-Arvizu analyzed the data, reviewed drafts of the paper.

David A. Shoham analyzed the data, wrote the paper, reviewed drafts of the paper.

Guichan Cao analyzed the data, prepared figures and/or tables, reviewed drafts of the paper.

Soren Brage, Ulf Ekelund and Richard S. Cooper conceived and designed the experiments, wrote the paper, reviewed drafts of the paper.

The following information was supplied relating to ethical approvals (i.e., approving body and any reference numbers):

This study was approved by the Institutional Review Board (IRB) of Loyola University Chicago, IL, USA; the Committee on Human Research Publication and Ethics of Kwame Nkrumah University of Science and Technology, Kumasi, Ghana; the Health Sciences Faculty Research Ethics Committee of the University of Cape Town, South Africa; the Board for Ethics and Clinical Research of the University of Lausanne, Switzerland; the Research Ethics Committee of the Ministry of Health in Seychelles; the Ethics Committee of the University of the West Indies, Kingston, Jamaica; and the Health Sciences IRB of the University of Wisconsin, Madison, WI, USA.

The following information was supplied regarding data availability:

The raw data has been supplied as a Supplemental File.

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
