# Peer review of "Accelerometer-measured physical activity is not associated with two-year weight change in African-origin adults from five diverse populations"

_PeerJ, doi:10.7717/peerj.2902_

## Round 0.1 · original submission · Major Revisions

Please pay special attention to the comments of Reviewer 1 who feels that the article currently fails to account for all possible effects,

·

Basic reporting

No comments

Experimental design

I am not sure of these five cohorts are similar enough to be directly used in the pooled analysis

Validity of the findings

No comments

Additional comments

The authors/affil list does not match in the first three pages? The study is interesting, but seems to lack two main ingredients – the cohort effect (where one would assume that the average weight increases with age) and also a decent food frequency / diet composition indication. Ignoring these introduces a substantial amount of uncontrolled effects, which could take over a majority of variance. It would have been great if this effect could have been controlled for, as it would then be possible to attribute a proportion of variance to both effects in a single analytic model. The use of analytic model ensures optimal account for the diverse study design and follow-up features. I would have excluded all pregnant women, as their proportion of weight variance increases potentially with pregnancy, which is a strong modifying effect for weight. Also, I did not see any adjustment to major changes in lifestyle, which could have also affected the result. I am also not sure how similar these five cohorts are in terms of pooled analysis vs meta-analytic approach (having in mind wildly diverse ethnical backgrounds of African American people, their cultures and dietary patterns). The study does contribute to understanding of the effects of PA, but it somehow fails to provide a decisive answer.

·

Basic reporting

The manuscript is clear, unambiguous, and adheres to all PeerJ policies. The figures are relevant, well labeled and appropriately described, and raw data are supplied.

Experimental design

The manuscript is within the scope, and the research question is relevant.
The research was done rigorously and followed ethical standards, and is described in sufficient detail to be reproducible.

Validity of the findings

The findings are valid and supported by the supplied dataset, but the authors should look at body composition (which they measured and have data available) and not just/only BMI. BMI and total weight are not fine-enough tools to dissect and use for (eventual) evidence-based public health measures.
Recent concepts such as the obesity paradox, metabolically healthy obesity, and metabolically obese normal weight should be discussed after revisiting the available data for fat free mass and fat mass.
Also, I think it is important to revisit the original data and observe some (unexpected) values for some of the participants - some participants height changed drastically at their third visit (eg. check participants 3001 - 2 cm; 3004 - 3 cm; 3012 - 2 cm; 3014 - 5 cm!...).

Additional comments

The authors should adequately comment and addend the manuscript on the topics of:
1) extrapolation of PA during (only) 8 days and interpretation that this was representative for the entire study period (especially taking into account different times in the year for different locales);
2) the fact that only/mostly subjects of predominantly African descent were involved in the study and how that may (or may not) influence the conclusions;
3) their choice of using BMI and total body weight change, and not the percentage of lean muscle (fat free mass) or fat mass (percentage of fat) that the authors measured; and how those changed over the examined period - as BMI is not an ideal measurement of true body composition (especially for very physically active individuals).
4) Minor spelling mistakes - line 47 (tender - tended); line 49 Neither - nor; 104 are - were; 123 using a - using the; 197 who?)

Reviewer 3 ·

Basic reporting

This article explores the important issue of how physical activity (PA) affects weight change in diverse populations. The authors provide a generally well-written article with a few grammatical errors, but the Introduction and Discussion sections lack some essential elements to allow readers a thoughtful review of potential reasons as to why physical activity was not associated with weight gain. The authors need to state the main objective of the study in the Introduction section. Other than these issues, the tables and figures allow readers to follow clearly the reported results and the authors interpret the results appropriately. The strengths of this study include the objective assessment of PA and the cross-cultural comparison of results. Once revised, this manuscript should be of interest to public health researchers and health professionals. Please find detailed suggestions below divided by section.

Experimental design

Abstract
1) Please change the word ‘tender’ in line 18 page 2 to ‘tended’. Also on line 19 please insert the word ‘neither’ at the beginning of the sentence that starts with ‘Baseline…’ and on line 20 please change ‘significant’ to ‘significantly’.
Introduction
1) The authors need to state the objectives of the current study as a last paragraph of this section. After describing the drawbacks of previous studies on the relationship between PA and weight gain, I suggest the authors state the purpose of this study. Perhaps a sentence stating: ‘The purpose of this study was to objectively assess whether PA is associated with weight change in a cohort of diverse populations, or a similar statement. ‘This is currently missing from this manuscript.
2) Please move the third paragraph on p. 4 lines 53-65 to the Materials and Methods section under the suggested subheading ‘Participants and Data Collection’. Please incorporate this whole paragraph describing the data collection along with inclusion and exclusion criteria into the Materials and Methods section.
Materials and Methods
1) Please merge data about the participants and the Modeling the Epidemiologic Transition Study (METS) from the last paragraph of the Introduction section with the participant recruitment data in the first paragraph of this section.

Validity of the findings

Materials and Methods
1) Please merge data about the participants and the Modeling the Epidemiologic Transition Study (METS) from the last paragraph of the Introduction section with the participant recruitment data in the first paragraph of this section.
Discussion and Conclusion
1) The authors should briefly discuss more factors involved in the very complex relationship between obesity and physical activity. For instance, not just food consumption plays a role in weight gain, but interpersonal differences and perceptions, the built environment, and social-environmental factors, for example.
2) Suggestions for minor changes: on p. 13, line 292 please change the word ‘it’ to ‘in’. On line 294, please delete ‘of’ after ‘… yearly weight change’. On p. 5 line 295, please change the word ‘was’ to ‘were’.

---

## Round 0.2 · accepted · Accept

You have addressed all relevant issues raised during the review process.

·

Basic reporting

The improvements made to the manuscript are fine, and I recommend it for publication

Experimental design

OK

Validity of the findings

OK

Additional comments

OK